# Seasonal forcing and waning immunity drive the sub-annual periodicity of the COVID-19 epidemic

**Ilan N. Rubin**⦿*, **Mary Bushman, Marc Lipsitch, William P. Hanage**

Center for Communicable Disease Dynamics, Department of Epidemiology, Harvard T.H. Chan School of Public Health, Boston, Massachusetts, United States of America

* ilan.n.rubin@gmail.com

## Abstract

Seasonal trends in infectious diseases are shaped by climatic and social factors, with many respiratory viruses peaking in winter. However, the seasonality of COVID-19 remains in dispute, with significant waves of cases across the United States occurring in both winter and summer. Using wavelet analysis of COVID-19 cases during the pandemic period, we find that the periodicity of epidemic COVID-19 varies markedly across the U.S. and correlates with winter temperatures, indicating seasonal forcing. However, seasonal forcing alone cannot explain the pattern of multiple waves per year that has been so characteristic of COVID-19. Using a modified SIRS model that allows specification of the tempo of waning immunity, we show that specific forms of non-durable immunity can sufficiently explain the sub-annual waves characteristic of the COVID-19 epidemic.

## Author summary

Many respiratory viral illnesses are known to have seasonal trends, with higher transmission and cases in the winter. Using wavelet analysis, we show that the period of reoccurring waves of COVID-19 varies across the United States and is correlated to winter temperatures. Areas of the United States with colder winters generally experienced the winter wave of COVID-19 cases expected from the classical pattern of seasonally forced diseases. However, areas with warmer winter climates regularly experienced multiple waves of new infections each year during the pandemic period. While the evolution of SARS-CoV-2 unarguably played a large part in determining the severity of the variant waves, it is unlikely to be a good explanation for these climatic trends. Using a compartment-style epidemic model, we also explain how waning immunity following the disease can explain the multiple waves per year that have been a uniquely disruptive characteristic of the COVID-19 pandemic.

**Data availability statement:** Code for both the spectral analysis and model simulations is available at https://github.com/ilanrubin/COVID-19-Seasonality.

**Funding:** This work was funded by a grant from the National Institutes of Health, SeroNet cooperative agreement U01CA261277 (INR, MB, ML, WH), the National Institutes of Health grant T32AI007535 (INR), and by CDC contract 200-2016-91779 (INR, MB, ML, WH). The funders had no role in study design, data collection and analysis, decision to publish, or preparation of the manuscript.

**Competing interests:** I have read the journal's policy and the authors of this manuscript have the following competing interests. WPH is an advisor to Biobot Analytics, and a consultant to Shionogi Inc.

## Introduction

Much of the scientific and public discourse surrounding the COVID-19 epidemic and its waves has focused on the named "variants of concern" (VoC) [1,2] that have dominated at different times. While VoCs have played an unquestionably large role in the magnitude of the pandemic [3], the question of the timing of each wave is likely separate as we do not expect VoCs to emerge with a regular periodicity. The evolution of SARS-CoV-2 may therefore have obscured the degree to which COVID-19 shows evidence of seasonality, in common with other respiratory viruses.

Many infectious diseases show pronounced seasonal patterns, producing waves of cases at particular times each year. Such seasonality is important for practical purposes in public health, allowing improved forecasting of heightened healthcare demand. Agents that transmit via the respiratory route are typically associated with a peak in the winter months, for reasons that may be due to climatic conditions and/or changing contact patterns during that period [4]. In the case of the COVID-19 epidemic, waves of transmission have occurred not only in the winter, but also during the spring or summer, and have thus been out of sync with the classical pattern of respiratory viral pathogens [5]. Untangling the seasonality of the COVID-19 epidemic from patterns in viral evolution, population immunity, the timing of government mitigation efforts, and epidemic stochasticity is a difficult yet imperative, task.

In the United States, multiple waves of infection have been documented each year since the start of 2020. Epidemic severity has tended to alternate between the northern and southern United States with evidence for a strong summer wave in the southern United States [6] as well as a negative temporal correlation between the number of cases and absolute humidity [7]. This periodic regularity may be a separate phenomenon from the appearance of VoCs and so the precise patterns of seasonality, and thus their predictability going forward, remain to be determined.

While changes in case ascertainment make it hard to determine absolute incidence in the absence of large-scale community surveys (such as [8]), wavelet analysis offers an alternative approach to assess changing periodicity. Wavelet analysis is a spectral decomposition method similar to continuous Fourier transformation that is also able to handle non-stationary time series data and thus relevant in the pandemic context. This approach does not assume a single fixed periodicity and can therefore determine whether the oscillatory characteristic of the epidemic changes over time [9]. For example, Grenfell et al. showed how the periodicity of measles cases in London changed after the onset of vaccination [10]. Similar analyses of the COVID-19 epidemic for various countries around the world [11,12], have as yet analyzed time periods in which only one or two epidemic waves had taken place and therefore offer limited insight into the long-term trajectory of the disease and the relative impact of VoCs.

The United States offers a useful case study for how climate effects epidemic seasonality, having a range of very different climates together with readily available estimates of case counts. Here we use an exploratory wavelet analysis of the periodicity of COVID-19 waves in the United States using state and county level COVID-19 case estimates and determine how periodicity is correlated with measures of mobility, climatic, and demographic variables.

## Results

### COVID-19 displays both annual and sub-annual periodic signals

We applied wavelet decomposition to the logarithm of the 7-day rolling averages of cases per 100,000 individuals as compiled by the New York Times from January 21, 2020 through March 24, 2023, the entirety of the time they were reported [13]. We analyze the periodicity of COVID-19 cases over that time-period using wavelet analysis – a technique to resolve a continuous time series into time-varying periodic components.

Wavelets of new infections at the state level were generally very stationary (for an example cases time-series and wavelet, see Fig 1 showing the log cases, wavelet periodogram, and global wavelet spectrum for the state of Massachusetts), with the periodic signal often being comprised of two ridges – one roughly annual and one sub-annual (with a period in the range of 3–6 months). Within each state, this dynamic remained fairly constant across the entire testing period. Perhaps surprisingly, large-scale evolutionary events such as the first Omicron wave in late 2021 had little effect on each wavelet and the underlying periodicity of the infection dynamics. Instead, the evolutionary dynamics of SARS-CoV-2 and the immune evasion or increased transmission rate of the different VoCs seem to significantly affect the magnitude of the waves but otherwise fall within the periodic dynamics of the region.

As the wavelets are largely stable over this time period [e.g., Fig 1], we continue to analyze the global wavelet spectrum, which is calculated by averaging the wavelets over the entire time period and is analogous to a Fourier spectrum. The global wavelet is a distribution of the average periodic components that make up the time-series and is a simple summary of the periodicity of COVID-19 cases over the entire time period in question.

The state-level global wavelet spectra are relatively consistent across U.S. states and territories and generally show two major components [Fig 2A]. The first has a period of approximately 365 days, as one would expect for a disease with classical winter waves of infections. The second major component is sub-annual and generally peaks between 120 and 180 days depending on the state and is consistent with the pattern of multiple large waves of transmission that occurred each year from 2020-2023 for most of the United States.

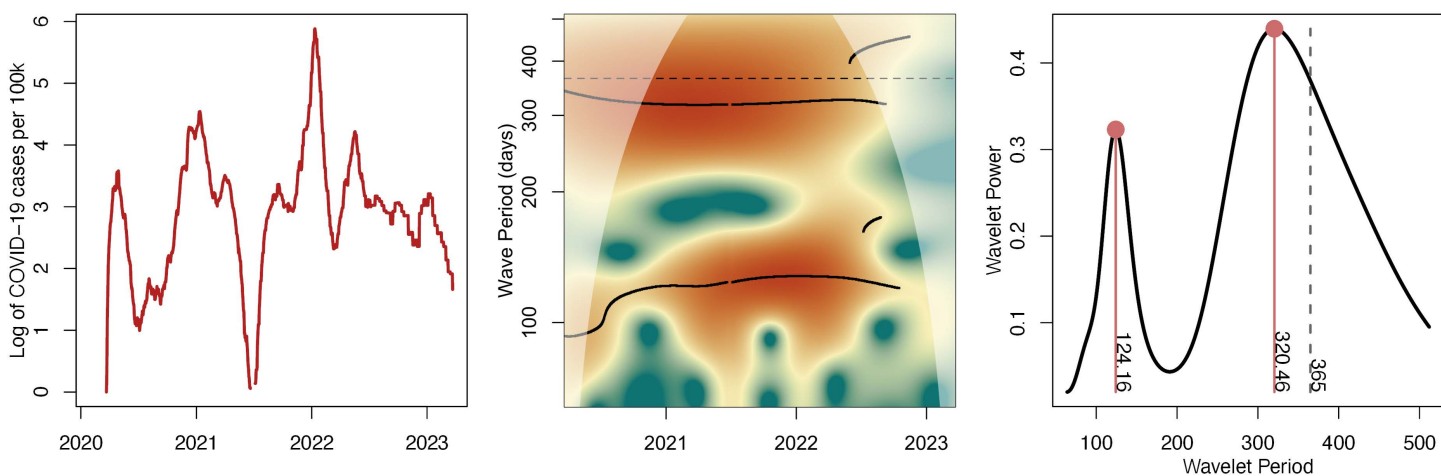

**Fig 1. Massachusetts COVID-19 cases and periodicity.** (A) The log COVID-19 cases per 100,000 residents in Massachusetts as estimate by the New York Times [13]. (B) A wavelet periodogram of the log COVID-19 cases per 100k in Massachusetts. Color represents the power of the wavelet. The area with lighter coloring is outside the cone of influence and may be affected by edge effects. Significant ridges in the spectrum are shown as black lines. The two horizontal black lines represent stable components with periods of slightly less than a year and approximately four months. (C) The global wavelet spectrum for Massachusetts log cases per 100k, representing the average periodic components over the time-series.

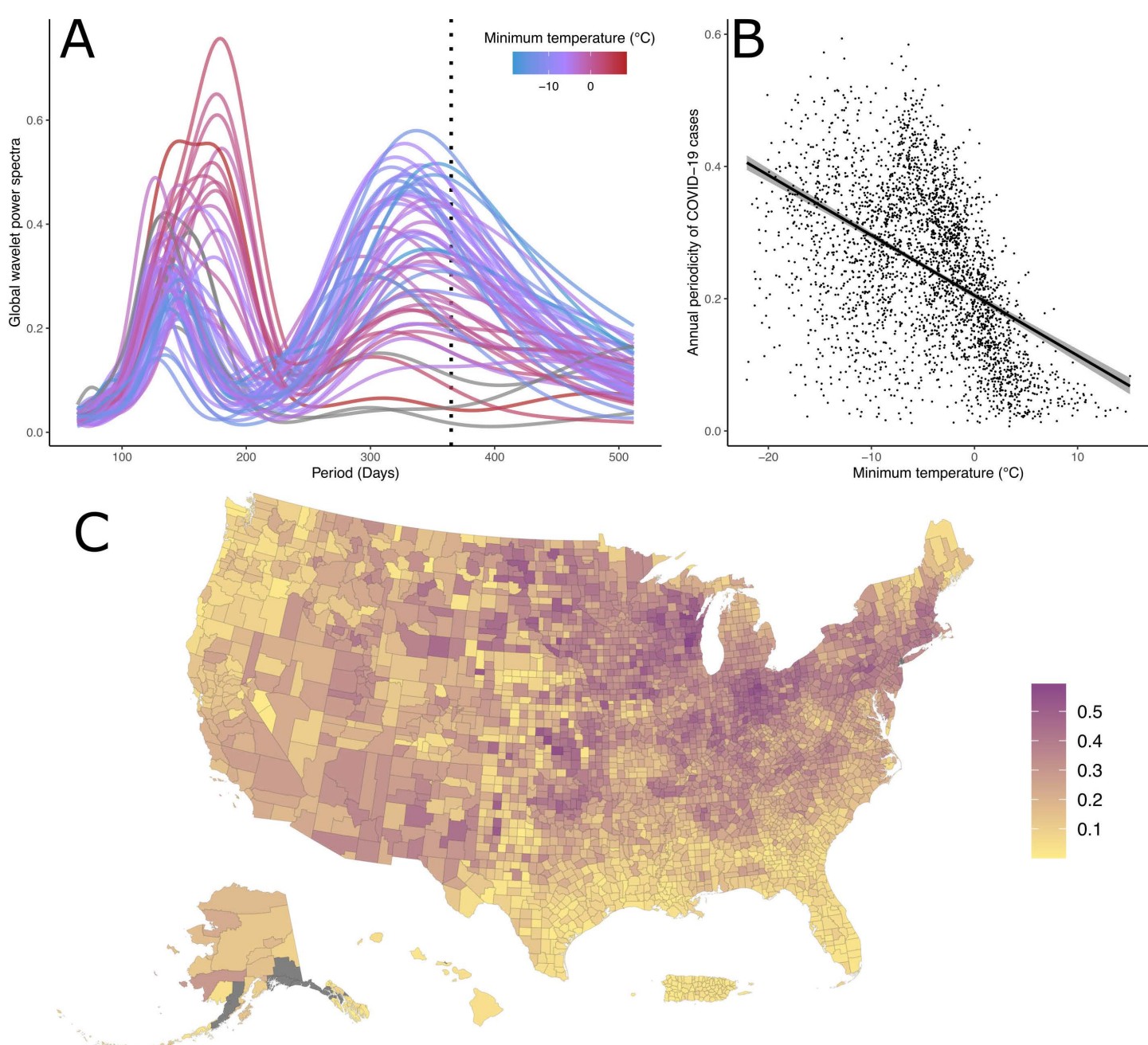

**Fig 2. COVID-19 periodicity across the United States.** Wavelet transformations were calculated for log of COVID-19 incidence for each U.S. state, Puerto Rico and the Virgin Islands. (A) Curves are colored by the minimum of the state-wide average monthly low temperature measured between 2020 and 2023. States with a warmer winter temperature are shown in red and cooler states in blue. Areas not included in NOAA's historical climate dataset (Alaska, Hawaii, Puerto Rico, and the Virgin Islands) are shown in grey. A dashed line is shown at 365 days. (B) COVID-19 annual component as a function of temperature seasonality for each county in the United States. The magnitude of the annual component was calculated as the power of the global wavelet spectrum at 365 days. A linear regression and its 95% confidence interval are also shown. (C) A map of all counties in the United States colored by the magnitude of their annual component. The base map for all county borders was sourced from U.S. Census Bureau's 2020 TIGER/Line Shapefiles [50].

## COVID-19 periodicity is correlated to temperature

States with warmer winters (as quantified here by the average over 2020–2023 of the lowest of the monthly minimum temperature each year according to NOAA's Monthly U.S. Climate Divisional Database [14]) were dominated by the sub-annual component and display little, if any, annual seasonality [Fig 2A]. Conversely, states with colder winter low temperatures have annual and sub-annual components of similar magnitude in their global wavelet spectra and display a comparatively larger annual signal than states with warmer winters. Further supporting this pattern, of the four states and territories (Alaska, Hawaii, Puerto Rico, and the Virgin Islands – shown in grey) that are not included in the NOAA Climate Divisional Database, three have tropical climates and show essentially no signature of annual periodicity, while Alaska has a larger annual component and very cold winters. This difference in epidemic periodicity can be seen in the case counts as Florida, Louisiana, and Georgia (the states with the warmest winter climates) experienced two relatively similarly sized waves a year, while Minnesota, North Dakota, and South Dakota (the states with the coldest climates) experienced a large winter wave each of the three years with no or much smaller off-peak waves [S4 Fig].

We extend this analysis to the county level and quantify the strength of the annual component as the power of the global wavelet spectrum at a period of 365 days. While this quantification is a relatively simple description of the seasonality of the epidemic in each county, it provides an intuitive way to summarize, in a single number, how strong the annual-scale periodicity of COVID-19 across the United States. In doing so, a clear pattern emerges. Counties in the southern and western United States, and particularly the southeast, show little annual periodicity, while counties in the Midwest and northeast of the United States experienced much more annual epidemics [Fig 2C]. Counties around the Rocky Mountains have a less distinct pattern yet are also often less populous and therefore likely more dominated by stochastic dynamics.

In concert with this visually apparent trend [Fig 2A], the power of the annual component for the COVID-19 epidemic across U.S. counties is significantly negatively correlated to temperature minimums (as defined by the average over 2020–2023 of the lowest monthly minimum temperature each year in the NOAA's Climate Divisional Database [14]) across U.S. counties [Table 1, Fig 2B]. Counties with colder winters were more likely to have experienced strongly annual COVID-19 epidemics, while counties with warmer winter temperatures were more likely to have epidemics dominated by sub-annual waves.

The maximum temperature of the county is also negatively correlated (to a lesser degree) with the strength of the annual component, a further link between colder climates and strongly annual COVID-19 epidemic dynamics [Table 1]. However, temperature variability (as defined by the difference between summer high and winter low temperatures) is positively correlated, possibly indicating that an annual epidemic cycle is associated with the overall extremity of the regional climate as well, rather than simple cold winters. A multivariable regression shows that the annual component is associated with both the minimum and maximum temperatures in a location (along with many of the other variables), further emphasizing the role of temperature in predicting stronger annual cycles [S1 Table].

## A model of sub-annual periodicity considering individual waning immunity

While the previously described wave decomposition analyses provided insight and intuition into the character of the periodicity of the COVID-19 epidemic, these analyses are non-mechanistic. Previous theory on seasonal forcing shows how epidemics can be forced into an annual cycle or longer multiennial cycles [10,15,16]. For instance, Dushoff et al. show how undetectably small changes in transmission rate over the course of an annual cycle can result in large oscillations in influenza incidence through dynamical resonance of the disease and seasonal dynamics [17]. However, this seasonal forcing theory largely pertains to epidemics with annual or longer cycles.

The classical SIRS model (Susceptible-Infectious-Recovered-Susceptible) can drive waves in the number of infected individuals through cycles of the depletion and replenishment of the number of susceptibles as immunity in the population wanes. However, the classical SIRS model does not consider partial or waning immunity, but absolute immunity that

**Table 1. Correlations between COVID-19 annual component and various climatic and socio-demographic variables.** Wavelet analyses were calculated for the log of COVID-19 cases until March 2023 as estimated by the New York Times for each county (or county equivalent) in the United States. Regressions were calculated for the global wavelet transformation (average of the wavelet transformation over time) for a period of 365 days as a function of various climatic and socio-demographic variables. Positive slopes indicate that an increase in the variable in questions is correlated to an increase in the power of the annual component of COVID-19 in that county. Due to the large sample sizes, all correlations were statistically significant. Variables with $r^2 > 0.1$ are bolded.

| Variable | Pearson's correlation | $r^2$ | source |
|---|---|---|---|
| **Minimum temperature** | −0.432 | **0.228** | [14] |
| **Maximum temperature** | −0.326 | **0.114** | [14] |
| **Difference of max and min temperature** | 0.354 | **0.166** | [14] |
| Average monthly precipitation | −0.113 | $2.89 \cdot 10^{-2}$ | [14] |
| Log population size | 0.207 | $2.09 \cdot 10^{-2}$ | [45] |
| % Population age 65+ | −0.130 | $9.02 \cdot 10^{-3}$ | [45] |
| % Population under 200% of the poverty line | −0.291 | $9.05 \cdot 10^{-2}$ | [45] |
| **% Population Insured** | 0.391 | **0.157** | [45] |
| Log population density | 0.157 | $8.86 \cdot 10^{-3}$ | [45,46] |
| Log geographic area | $2.29 \cdot 10^{-2}$ | $1.63 \cdot 10^{-3}$ | [46] |
| Republican vote %, 2020 presidential election | $3.56 \cdot 10^{-2}$ | $5.75 \cdot 10^{-3}$ | [48] |
| Google mobility periodicity (retail) | 0.173 | $2.99 \cdot 10^{-2}$ | [47] |
| Google mobility periodicity (workplace) | −0.117 | $1.37 \cdot 10^{-2}$ | [47] |
| Log total number of COVID-19 cases | 0.199 | $2.66 \cdot 10^{-2}$ | [13] |
| **HHS region (ANOVA)** | | **0.241** | [47] |

wanes immediately (with the time persons are immune distributed exponentially in the population) [18]. Under these classical assumptions, while the inter-epidemic period can be controlled as a function of the average length of time an individual is immune, epidemic waves are always damped oscillations (decrease in size from one wave to the next) and are thus not maintained over long periods [19]. The first years of the COVID-19 pandemic were characterized by a regularly occurring series of waves of new infections of equal or successively greater magnitude, a pattern that cannot be explained by the classical SIRS model.

To provide intuition into possible mechanisms that could be driving this more unusual sustained sub-annual periodic behavior, we introduce a generalization of the classic SIRS epidemic model that includes both partial immunity that wanes over time as well as seasonal forcing [Fig 3]. Immunity to reinfection is known to wane [20,21] and seasonal forcing is a significant periodic driver in other respiratory pathogens [15,22]. Here we prove whether one or both of these mechanism can explain the sub-annual periodicity we have described.

The model can be described as a system of partial differential equations with Susceptible ($S$), infectious ($I$), and Recovered ($R$) classes that are a function of time ($t$) and time since recovery from infectiousness ($a$). Individuals can become infected from both the Susceptible and Recovered classes with immunity from infection for the Recovered class described by a function $\omega(a)$. While this function can take any form, here we use a functional form based on the cumulative distribution function for a Weibull distribution that allows for easy tuning between exponentially and more stepwise (sigmoidal) decaying immune waning functions [e.g., see Fig 3 and S7 Fig]. The immunity waning function is governed by three parameters: $k$ is the shape parameter that controls the steepness of the waning, $\lambda$ represents the "half-life" and controls the timing, and $\omega_\infty$ is the asymptote or amount of immunity maintained for perpetuity. For the equations and a more detailed discussion of the model, see materials and methods.

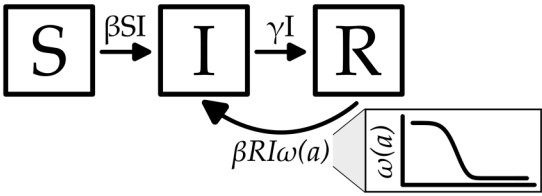

**Fig 3. SIR model with waning immunity compartment diagram.** A representation of the proposed SIR model with waning immunity. *S*, *I*, and *R* represent the susceptible, infected, and recovered states respectively and are all functions of time, *t*. $\beta$ is the transmission rate and may be either a static variable or a function of time, *t*, while $\gamma$ is the rate of recovery from illness. In the classical SIRS model, recovered individuals lose all immunity and return to the susceptible compartment after some time. In the model presented, individuals remain "recovered" for perpetuity but over time can be reinfected as a function of waning immunity $\omega(a)$, where *a* is the time since recovery. In this model *S* represents an initial state made up of individuals who have never been infected and have no immunity.

For computational simplicity, we simulated an individual-based version of the SIR model with waning immunity and seasonal forcing. We ran 125,000 simulations using Latin hypercube sampling [23] to sample the three immune waning parameters with different amounts of annual seasonal forcing as part of the transmission rate. The simulations were then analyzed based on the periodicity of newly infected individuals as defined by the average inter-epidemic period between waves in order to understand which immunity regimes generate sub-annual epidemic periodicity that is maintained over a relatively long period of time (i.e., effectively undamped oscillations) with and without seasonal forcing. Annual periodicity is defined as any simulation with an average inter-epidemic period of between 9 and 18 months and sub-annual of less than 9 months, both requiring more than 5 total waves during the 10 simulation years to filter out strongly damped oscillations. Simulations that did not result in either annual or sub-annual waves either reached a stable equilibrium endemic state or resulted in waves less frequent than every 18 months.

The model is able to produce sub-annual waves when three conditions are met: there is little long-term immunity ($\omega_\infty < 0.5$), immunity wanes relatively sharply ($k > 1.5$), and half-life time of immunity is short ($\lambda < 200$ days) [S7 Fig]. In practical terms, this represents waning functions that have a sigmoidal shape and thus a period of strong immunity, a transition period with partial immunity that occurs with about half a year or less, and very little immunity in the long-term. This durability is roughly in line with epidemiological estimates of the durability of immunity to SARS-CoV-2 [20,21,24].

In general, the shape parameter of the waning function ($k$) and the amount of long-term immunity ($\omega_\infty$) determine whether stable periodicity is possible, while the frequency of that periodicity is largely a function of the half-life of the immunity ($\lambda$). For example simulation trajectories and immune waning functions, see S7 Fig. Adding seasonal forcing to the model can create more variable and chaotic dynamics than the damped or stable waves generated from just waning immunity [for example trajectories with seasonal forcing see S10 Fig] as the two periodic generating mechanisms (seasonal forcing and waning immunity) may resonate or interfere with each other.

The only parameters varied in these simulations relate to immunity, while leaving transmission rates and infection times fixed. As a result, annual attack rates are largely determined by how many waves occur each year, rather than the magnitude of each wave. Diseases with recurrent waves within a single year (e.g., COVID-19) are hence expected to have a higher annual attack rate compared to those that follow the classical annual periodic pattern as the disease runs through the population multiple times [25]. Thus, in these simulations, annual attack rates are largely correlated to the the half-life of immunity ($\lambda$), the immune parameter most correlated with the frequency of periodicity [S8 Fig].

Perhaps most importantly, including annual seasonal forcing on the transmission rate had little to no effect on the generation of sub-annual periodicity and instead only increased the likelihood of annual waves [Fig 4]. This is still true if only the first four years of the simulation are considered [S9 Fig] to match the period of COVID-19 case counts we analyzed in the previous section. Thus, while seasonal forcing undoubtedly would impact the magnitude and timing of the epidemic

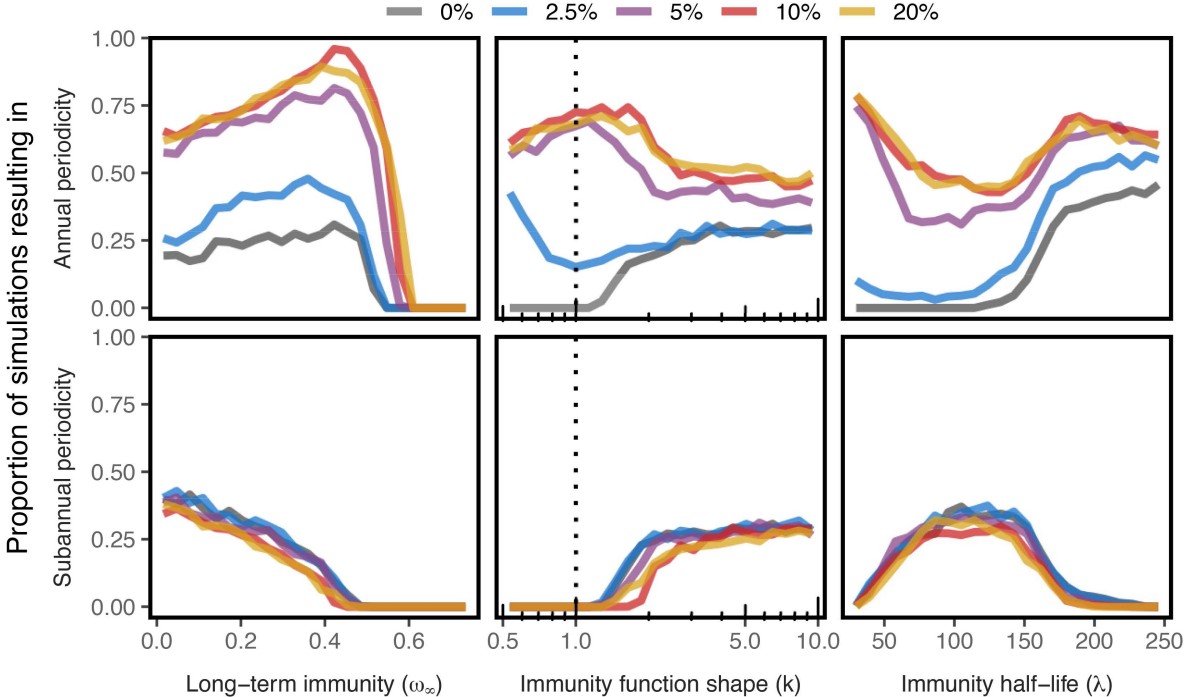

**Fig 4. Seasonal forcing does not markedly affect sub-annual periodicity in simulated epidemics.** We ran 125,000 simulations of the individual-based implementation of the SIRS model with individually waning immunity, with 25,000 simulations for each of five different parameterizations of seasonal forcing (shown in color with $\theta$ = 2.5%, 5%, 10%, and 20%, or when the peak transmission rate is $\theta$% higher transmission than at its lowest). Immune parameters were chosen randomly with Latin Hypercube Sampling for each seasonality regime ($\omega_\infty \in (0, 0.75)$, $k \in (0.5, 10)$, and distributed logarithmically $\lambda \in (25, 250)$). Annual periodicity is defined as any simulation with an inter-epidemic period of between 9 and 18 months and sub-annual of less than 9 months, both requiring more than 5 total waves during the 10 simulation years to filter out strongly damped oscillations. Simulations that did not result in either annual or sub-annual waves either reached a stable equilibrium endemic state or resulted in waves less frequent than every 18 months. The probability of an annually recurring epidemic generally increases with increasing seasonal forcing, dependent on the immunity characteristics. However, the presence of seasonal forcing has little to no effect on driving sub-annually reoccurring epidemics. The dashed lines represent exponentially waning immunity, a good proxy for the classic SIRS model, and where undamped sub-annual waves are never generated. For an epidemic to be periodic at all immunity must wane sharply ($k \gtrsim 1.5$) and the long-term immunity must be marginal ($\omega_\infty \lesssim 0.5$).

waves, likely in a chaotic way due to the resonance and synchrony of the two processes [26,27], it is unlikely to change the importance of non-durable immunity in generating the such rapidly reoccurring epidemic waves unusual to the COVID-19 epidemic.

However, simulations with increased seasonal forcing do result in an increased probability of annual periodicity, with no accompanying reduction in the chance of sub-annual periodicity. This increase in annual periodicity possibly provides an explanation for the temperature-mediated periodic patterns we described earlier in which areas of the United States with colder winters – and thus likely stronger seasonal forcing – showed a dominate annual signal in COVID-19 cases.

## Discussion

Here we presented preliminary evidence for seasonality of COVID-19 in the first few years following the emergence of SARS-CoV-2. We have demonstrated that the periodicity of pandemic COVID-19 can be characterized by annual and sub-annual dominant components, with the degree of sub-annual periodicity correlated mostly strongly to winter low temperatures. Previous work on early pandemic dynamics has indicated higher transmission rates during the winter through direct correlations of cold temperatures with case counts [7,28] and that SARS-CoV-2 degrades more slowly in low

temperatures and extreme relative humidity [29], a possible mechanism driving this periodic variability. However, the small a negative correlation with high temperatures in the summer may indicate that an annual epidemic cycle is associated with the overall extremity of the regional climate, rather than simply the cold winters.

In contrast with temperature, most socio-demographic characteristics of the county had little to no association with annual periodicity. Population size, population density, elderly population, geographic area of the county, and the political leaning of the county show little correlation to the COVID-19 epidemic periodicity [Table 1, S6 Fig]. Thus, while the political partisanship of a county is known to be significantly correlated with the magnitude of the epidemic in that county [30], the timing of their waves was seemingly uncorrelated. We also, conducted similar wavelet analyses of Google mobility data (specifically their "retail and recreation" and "workplace" variables), that is both visually and intuitively periodic on an annual cycle (people move around more in the summer than winter). However, the mobility of the population explained under 3% of the variation in epidemic periodicity, compared to the more than 20% that is explained by temperature seasonality.

Other variables that show relatively high correlation with the annual periodicity of the epidemic may be explained by their association with climate. For instance, of the ten Health and Human Services regions, the region of states around the Great Lakes (Region 5) has a significantly higher level of annual periodicity [S6 Fig], a pattern that is consistent with temperature minimum. Similarly, the correlation with the percentage of the population with health insurance can be largely explained by low rates of insurance in Texas and Florida [S5 Fig], two states with comparatively warmer climates.

While this work cannot establish the precise mechanisms underlying the periodicity we observe, we note that the regular timing of the winter peak each year [S3 Fig] is generally consistent with the winter holiday period even in places with little evidence of an annual component (e.g., Florida) [S4 Fig] and perhaps consistent with the increased mixing and changing contact patterns over the holiday period. However, holiday travel is relatively consistent across the U.S., yet COVID-19 periodicity is not and is instead correlated with temperature, supporting the possible climatic mechanism to seasonal forcing. Further comparisons with other regions around the world with different holiday schedules and climate variability may provide further evidence for this proposition.

Most importantly, the positive correlation between the extent of climatic variability and annual periodicity is a strong indication that the observed periodicity is not simply due to chance arrivals of new SARS-CoV-2 strains at approximately annual intervals, and instead suggests that some climate-related drivers (including behavioral consequences) are an important driver of annual seasonality. A prediction of the hypothesis that seasonal forcing drives the annual periodicity is that this periodicity should be stronger where seasonality is stronger. Our findings are consistent with this finding, and would be hard to explain parsimoniously if the annual period were a byproduct of chance timing of introduction of new variants.

The evolution of SARS-CoV-2 has unarguably played a large role in the magnitude and duration of each wave of infections, yet evolution depends on the inherently stochastic nature of mutation while seasonality does not. Nor is it easy to explain how evolution alone can produce periodic behavior that is associated with climate (though see [31,32] for a possible theoretic basis for evolution induced waves). The regular timing of the waves that we have described implies a more deterministic mechanism (such as immune dynamics or seasonal forcing) has been a necessary driver of the periodic dynamics. While previous work finds that short-duration infections, continuous antigenic drift, and complete cross-immunity between strains can generate periodic waves of infections [31,32], here we show short-term immunity that wanes quickly is a sufficient condition for regular sub-annual periodicity, independent of multi-strain dynamics (see [33] for a more in-depth discussion of reinfection rates as function of recovery time).

Moreover, antigenic evolution effectively accelerates the rate with which the immunity of the population wanes as replacing strains out-compete their predecessors, and so the emergence of a VoC able to invade existing neutralizing antibodies has a similar effect to a sharply declining waning immunity function. Therefore, the effects of continual evolution and a waning immune response to a specific strain are difficult to practically separate and measure in natural populations.

An individual's "effective immunity" does not only consider their individual immune response, but also the whether the individual immune response is well matched to the virus that is circulating at that time. Thus, we are not suggesting individual immune response dynamics instead of evolution as the sole mechanism of periodic dynamics, but that the recruitment of susceptible individuals – whether through antigenic drift or the waning of an individual immune response (or some other demographic process) – is a necessary component of periodically reoccurring epidemics and that a sharply waning "effective" immunity is sufficient to explain the sub-annual periodicity we have previously described.

Previous work has examined periodic disease dynamics that are caused by or interacts with the repeated turnover or coexistence of viral strains [34,35]. Continued theoretical work aimed to understanding how large-scale evolutionary change in viral antigens – such as that which occurred in the BA.1, BA.2, and BA.2.86 variants [36,37] – interacts with the immunity-mediated sub-annual periodic dynamics described here is still warranted.

Just as evolution or waning immunity are unlikely to be correlated with climate, seasonality alone cannot explain the sub-annual periodicity we have shown, but it can emphasize an annual component [Fig 4. The stronger signal for annual periodicity we see in regions of the United States with colder winters may therefore be the result of these factors operating together, with waning immunity driving the sub-annual dynamics and strong seasonal forcing masking it in favor of a more annual epidemic in places with cold winters.

## Conclusions

While the patterns reported in this work have been consistent throughout the study period, the accumulation and loss of immunity, unforeseeable and stochastic viral evolution, and unpredictable nature of complex and possibly chaotic wave dynamics make detailed forecasting a challenge. While some have postulated that COVID-19 will inevitably transition to annual winter waves [22] that are a classical characteristic of other respiratory viral pathogens in the endemic state [15,17,22,38,39], we have shown here that this transition did not take place during the period of study, though annual waves have been a key part of the dynamics, especially in colder climates. Continued surveillance will be essential to monitor if these dynamics have stabilized into a predictable cycle or continue to evolve. As the immediate threat of the pandemic has dwindled, mild cases are likely to remain under-counted and present a challenge. While wastewater surveillance emerged during the pandemic as a new means to track disease trends in close to real time [40,41], it has not been universally applied and has remaining theoretical challenges as a direct replacement for high-quality estimates of disease incidence. Despite these challenges, more current wastewater surveillance data suggests that the patterns of regular winter waves of new cases across the United States and less consistent "off cycle" wave may still be relevant [42], reinforcing the importance of understanding the periodicity of the disease as a tool for predicting dynamics going forward. Nevertheless, a thorough analysis of wastewater surveillance data collected from diverse regions going forward would allow us to understand if the patterns observed here persist in the coming years.

We have shown evidence for distinct patterns of seasonality in COVID-19 cases across the U.S., correlated with climate. While evolution is known to have played a large role in the transmission and immune escape of SARS-CoV-2 [3], here we show that the periodicity of the COVID-19 epidemic remained relatively stable during the pandemic period despite large evolutionary events and is correlated to temperature lows. In warmer regions circulation throughout the year is marked by multiple sub-annual waves occurring every 3–4 months, while in regions with colder winters, an annual component predominates. We also show that annual seasonal forcing alone is unable to explain sub-annual waves but can produce such dynamics in concert with another periodic generating mechanism such as the steeply waning, short-term convalescent immunity proposed here. Although we are not able to disentangle exactly how viral evolution combines with the waning human immune response, we note that antigenic evolution produces accelerated waning. Improved understanding of how seasonal forcing and viral evolution interact with quickly waning immunity will permit future extensions of the classical SIRS model to describe other periodic epidemics.

## Materials and methods

### Periodicity analysis of COVID-19 incidence

We generated wavelet transformations of COVID-19 incidence using the package *WaveletComp* [43] in R [44]. We analyzed state and county-level log transformed 7-day rolling averages of new cases per 100,000 individuals as compiled by the New York Times from January 21, 2020 through March 24, 2023 [13]. A 7-day rolling average was used, rather than raw case counts, to control for spurious daily incidence numbers due to the chaotic nature of case counting during the epidemic and adjustments in reporting of daily estimates to correct for errors in prior estimates. Only counties with more than 500 total COVID-19 cases were included in the analysis to eliminate the highly stochastic nature of epidemics in small populations.

We then analyzed the resulting power spectra in the context of historical climate data (monthly average temperature highs and lows compiled by the National Oceanic and Atmospheric Administration's Monthly U.S. Climate Divisional Database – NClimDiv) [14], population demographic information base on the 2020 United States Census and American Community Survey [45,46], Google's Community Mobility Reports [47], and results of the 2020 presidential election [48]. For more detailed information on data sources and methodology, please see below.

Global wavelet spectra are defined as the average of the wavelet spectra over the entire course of the time series. The strength of the annual component of a global wavelet spectrum is used to quantify the magnitude of annual-scale periodicity in the epidemic and defined as the value of the global wavelet spectrum at exactly 365 days.

### Model with waning immunity

While the previously described wave decomposition analyses examine the periodicity of the COVID-19 epidemic, these analyses are non-mechanistic. The classic model used to provide intuition into possible mechanisms that could be driving this sub-annual periodic behavior, is the SIRS (Susceptible-Infectious-Recovered-Susceptible) compartment model. In this model, the population is divided into and moves between the Susceptible ($S$), Infectious ($I$), and Recovered ($R$) classes. It is commonly written as the following set of three ordinary differential equations:

$$\frac{S(t)}{dt} = -\beta(t)I(t)S(t) + \omega R(t) \tag{1}$$

$$\frac{dI(t)}{dt} = \beta(t)I(t)S(t) - \gamma I(t) \tag{2}$$

$$\frac{dR(t)}{dt} = \gamma I(t) - \omega R(t) \tag{3}$$

where $\beta$ is the transmission rate, $\gamma$ is the recovery rate, and $\omega$ is the rate of loss of immunity. Immunity in the SIRS model is completely sterilizing and the time individuals are immune is exponentially distributed with a mean of the inverse of the rate individuals revert to susceptible (i.e., the mean time immune = $\frac{1}{\omega}$). Periodicity in the SIRS model is driven by a return of individuals from the Recovered (aka immune) class to being susceptible.

Here we introduce a generalization of the classic SIRS epidemic model that includes individual partial immunity that wanes over time. This model can be described as a system of partial differential equations with the same Susceptible ($S$), infectious ($I$), and Recovered ($R$) classes, yet here they are a function of both time ($t$) and time since recovery ($a$). Individuals can become infected from both the Susceptible and Recovered classes with immunity from infection for the Recovered class described by a function $\omega(a)$. The proposed model is as follows:

$$\frac{S(t)}{dt} = -\beta(t)I(t)S(t)$$

(4)

$$\frac{dI(t)}{dt} = \beta(t)I(t)\left(S(t) + \int_0^\infty R(t,a)\omega(a)\,da\right) - \gamma I(t)$$

(5)

$$\frac{\partial R(t,a)}{\partial t} + \frac{\partial R(t,a)}{\partial a} = -\beta(t)I(t)R(t,a)\omega(a)$$

(6)

with the following boundary condition:

$$R(t,0) = \gamma I(t)$$

(7)

$\gamma$ represents the rate of recovery from the infected to the recovered, or immune, class $\beta(t)$ is the transmission rate at time $t$. $\beta(t)$ can be a scalar such that the transmission rate is fixed and independent of time or season. Alternatively, annual seasonal forcing can be modeled as a sinusoidal transmission function $\beta(t) = \beta_0(1 + \theta\cos((2t/365 + s_\theta)\pi))$, where $\beta_0 \in \mathbb{R}^+$ is the baseline transmission rate, $\theta \in [0, 1]$, represents the amplitude of seasonal forcing and $s_\theta \in [0, 1)$ represents a phase shift parameter that governs the time during year that peak transmission occurs. Any other functional form can also be used for $\beta(t)$ in place of the sinusoidal forcing.

This model allows for explicit control of the shape of the individual immune waning function, $\omega(a \mid k, \lambda, \omega_\infty)$, that is a function of the time since recovery from a previous case of COVID-19 infection ($a$). The individual waning function can take any form. Here we propose and analyze the following function:

$$\omega(a \mid k, \lambda, \omega_\infty) = exp\left(-\left(\frac{a}{\lambda}log(2)^{1/k}\right)^k\right)(1 - \omega_\infty) + \omega_\infty$$

(8)

The proposed function is loosely based on the Weibull cumulative distribution function and allows for a large variety of biologically relevant scenarios with few parameters. The waning immunity function wanes from 1, indicating full immunity, to $\omega_\infty \in [0, 1]$ that represents the long-term immunity (when $\omega_\infty = 0$ immunity is eventually completely lost). The rate of waning governed by a shape parameter $k \in \mathbb{R}^+$ and a timing parameter $\lambda \in \mathbb{R}^+$. Large $k$ represent a waning function that decreases from full immunity to $\omega_\infty$ in a stepwise manner, while smaller $k$ represents more gradual waning. The timing parameter $\lambda$ represents the "immunity half-life," or the time when immunity is halfway between 1 and $\omega_\infty$. If $k = 1$ and $\omega_\infty = 0$ the waning function describes exponential waning with an exponential decay rate of $1/\lambda$.

For examples of what this waning immunity function can look like with different parameterizations and the resulting dynamics, please see S7 Fig.

**Individual-based simulations**

We simulated an individual-based approximation of the proposed PDE model with discrete time (time resolution of a day). This was done for computational and programmatic ease. Individual immunity is tracked deterministically for each individual who has recovered from symptomatic disease. The number of susceptible individuals exposed to the disease each day is calculated as a binomial distribution with probability $p = 1 - exp(-I * R_0 * \gamma * \beta(t)/N)$ The code is available in an online repository.

Parameterization of the fixed parameters was chosen to roughly mimic that of COVID-19. As a relatively simple compartment-based model, it is not intended to precisely fit actual epidemic curves, but rather as a toolbox able to

reproduce the general dynamics (e.g., sustained sub-annual periodicity) with a few, easily tunable parameters. Thus, the specific parameterization is intended to be feasible but not based on specific data. The most important parameters used in simulations are show in S2 Table.

The individual based implementation is stochastic in nature, but when run on a population size of 1,000,000, resulted in sufficiently deterministic-like dynamics. Each simulation with the chosen parameterization was run for 3650 days. In order to eliminate the effect of the stochastic extinction of the disease, the simulation is reseeded with a single newly infected individual anytime it goes extinct.

After the simulation was completed, we counted the peaks in number of infected individuals. Peaks were identified with a recursive algorithm and defined as any point that is a local maximum and has a 1% prominence (the number of infected individuals are at least 1% higher at the local maxima than the minimum point between it and the next closest peak) or the simulation start or finish if there is no peak between it and one end of the time series. This is a pragmatic definition of an epidemic peak, but one we feel has more relevance to real-world epidemic scenarios than analytical determinations based on the differential equations as it dismisses any deterministically generated epidemic peaks that are too small to have practical significance as well as local maxima caused by stochasticity.

Annual periodicity is defined as any simulation with an average inter-epidemic period of between 9 and 18 months and sub-annual of less than 9 months, both requiring more than 5 total waves during the 10 simulation years to filter out strongly damped oscillations. Simulations that did not result in either annual or sub-annual waves either reached a stable equilibrium endemic state or resulted in waves less frequent than every 18 months.

### COVID-19 case data

COVID-19 case counts were taken from the New York Times' COVID-19 case estimates [13]. Cases were converted to cases per 100,000 individuals to control for population size between municipalities. The cases per 100k were then log transformed before spectral decomposition analysis to limit the out-sized effect of the initial Omicron wave in the winter of 2021–2022 and to emphasize the timing, rather than the magnitude of the waves.

### Climate data

Historical temperature and precipitation data for each state and county were obtained from the National Atmospheric and Oceanic Administration's National Centers for Environmental Information (NOAA NCEI) nClimGrid historical estimates. Area estimates for both the county and state level were calculated by NOAA NCEI as the mean of all grid point within the region in question, weighted by the cosine of the latitude of each grid point [49] and are not weighted by population.

### Population demographic data

We obtain estimates of the population size, elderly population, population in poverty, and population insured for each county from the 2020 U.S. Census American Community Survey [45,46].

### Mobility data

To estimate changes in mobility we use Google's mobility dataset [47]. We specifically use the "workplaces" and "retail and recreation" mobility variables as the two variables with the least sparse reporting. The mobility variables are presented as a percentage change compared to the baseline and "show how visits and length of stay and different places change compared to the baseline" [47]. The baseline is calculated during the 5-week period from January 3, 2020 through February 6, 2020.

To analyze the periodicity of the mobility variables at each location we perform the same wavelet decomposition analysis that we do on the COVID-19 case data. Similarly to with COVID-19 cases, the annual periodicity of a

county's mobility is then defined as the magnitude of the global wavelet spectrum at exactly 365 days. To ensure a reasonable estimate of the periodic signal, only counties with at least 500 days of mobility data were included in this analysis.

Notably, mobility is not equivalent to interaction between individuals, so likely does not capture the extent of the changing social network structure and increased contacts during the holiday season each year.

### Election data

We define the political leaning of each county as the percentage of voters in each county that voted for a republican candidate during the 2020 presidential election. Election data is sourced from The New York Time's 2020 election dataset [13].

### Mapping data

The shapefiles used to create all maps were obtained from the U.S. Census Bureau's 2020 TIGER/Line Shapefiles [50] and mapped using the *tigris* software package [51] in R RCT2024.

### Supporting information

**S1 Fig. Wavelets of COVID-19 for the states with the coldest (a-d) and warmest (d-f) winters climates.** Wavelets the log of the number of cases of COVID-19 per 100k are shown for the three states with the coldest winter climates (Minnesota, North Dakota, and South Dakota) and the warmest winter climates (Florida, Georgia, and Louisiana). Black lines depict ridges in the wavelet. Wavelets were relatively stable throughout the period of study allowing for us to simplify the analysis to use the global wavelet transformations. Global wavelet transformations for all 50 states are shown in S2 Fig. Historical temperature data comes from NOAA's Monthly U.S. Climate Divisional Database. Colder regions tended to have larger winter waves of new infections and little if any disease in the summer. Conversely, the warmer regions had large summer waves of new infections each year that were of similar or even larger magnitude to their winter waves.
(EPS)

**S2 Fig. United States map of global wavelet transformations.** Wavelet transformations were calculated for the log of COVID-19 incidence for each U.S. state. A dashed line is shown at 365 days for ease of identifying the power of the annual component of the global wavelet transformation.
(EPS)

**S3 Fig. Histogram of the timing of peaks in COVID-19 cases by week of the year.** Peaks in log COVID-19 cases were calculated for all FIPS in the United States as estimated by the NY Times [13] and are defined as any local maximum in the log of COVID-19 cases per 100k individuals with a minimum prominence of $10^3$. The histogram shows the number of peaks across all FIPS and the entire timeseries available that occurred during a given week of the year. This shows the regularity of the January winter wave, with less regularly timed waves occurring in the spring or summer and in the early winter.
(EPS)

**S4 Fig. Cases of COVID-19 for the states with the coldest (a-d) and warmest (d-f) winter climates.** The number of cases of COVID-19 per 100k are shown for the same six states as shown in S1 Fig.
(EPS)

**S5 Fig. County-level map of variables considered across the United States.** Maps of each variable used in comparison to the strength of the annual component of the COVID-19 epidemic at the county level. Where data is not available the county is shown in grey. Population size, population density, and county geographic area are colored on a log scale.

For a detailed description of each variable and its source, please see the material and methods. The base map for all county borders was sourced from U.S. Census Bureau's 2020 TIGER/Line Shapefiles [50].
(TIFF)

**S6 Fig. Annual component of COVID-19 cases versus variables considered across the United States.** The annual component is defined as the value of the global wavelet transformation at a period of 365 days. Each point represents a county or county equivalent region across the United States and its territories. Lines represent a simple linear regression and it's 95% confidence interval. HHS regions are plotted using a box-and-whisker plot that show the 1st through 3rd quartiles. Because of the large sample sizes, all regressions are statistically significant. For a detailed description of each variable and its source, please see the material and methods.
(TIFF)

**S7 Fig. Example simulated dynamics and waning immunity functions.** Four example individual-based simulations and the waning immunity functions that produced the dynamics. The top figure in each pair shows the number of newly infected individuals as a fraction of the total population. Red dots represent infection peaks and blue dots represent infection troughs as determined by our described algorithm. The bottom figure in each pair shows the individual waning immunity functions. Parameterization of each simulation are as follows: a) $\omega_\infty = 4.1 \cdot 10^{-4}$, $k = 5.4$, $\lambda = 144.8$; b) $\omega_\infty = 5.8 \cdot 10^{-4}$, $k = 1.9$, $\lambda = 234.0$; c) $\omega_\infty = 8.9 \cdot 10^{-6}$, $k = 1.4$, $\lambda = 166.6$; and d) $\omega_\infty = 0.5$, $k = 6.7$, $\lambda = 120.0$.
(EPS)

**S8 Fig. The attack rate compared to the half-life of immunity.** For a subset of 10,000 simulations, the cumulative attack rate (the total number of cases per capita per years of the simulation) are shown as a function of the half-life of immunity ($\lambda$). In simulations with very short immunity half-lives, many waves per year are possible, resulting in very high attack rates. Only simulations with $\omega_\infty < 0.5$ were used as to only consider parameter ranges that allow for periodicity. Seasonal forcing is fixed in all simulations as *theta* = 2.5%, the weakest forcing parameterization considered. All other parameters and ranges are unchanged from other simulations.
(EPS)

**S9 Fig. Seasonal forcing simulations over only four years.** The same 125,000 simulations of the individual-based implementation of the SIRS model with individually waning immunity as described in the main text, but only the first four years of the simulation are analyzed. The 25,000 simulations for each of five different parameterizations of seasonal forcing are shown in color ($\theta = 2.5\%, 5\%, 10\%$, and 20%, or when the peak transmission rate is $\theta\%$ higher transmission than at its lowest). Immune parameters were chosen randomly with Latin Hypercube Sampling for each seasonality regime ($\omega_\infty \in (0, 0.75)$, $k \in (0.5, 10)$, and distributed logarithmically $\lambda \in (25, 250)$). Annual periodicity is defined as any simulation with an inter-epidemic period of between 9 and 18 months and sub-annual of less than 9 months, both requiring more than 3 total waves during the 4 simulation years to filter out strongly damped oscillations. Simulations that did not result in either annual or sub-annual waves either reached a stable equilibrium endemic state or resulted in waves less frequent than every 18 months. The dashed lines represent exponentially waning immunity, a good proxy for the classic SIRS model, and where undamped sub-annual waves are never generated. The results are not meaningly different than when the longer 10 year time period is considered as shown in Fig 4.
(EPS)

**S10 Fig. Example simulated dynamics with seasonal forcing.** Four example individual-based simulations showing the complicated dynamics that can arise from the interaction of waning immunity and seasonal forcing. The top row shows two simulations that maintain sub-annual periodicity, though the magnitude of the waves is chaotic as a function of the seasonal forcing. The bottom row has two simulations in which the dynamics are "forced" onto an annual cycle. In C, the epidemic has sub-annual waves for the first few years, before settling into the annual cycle. This is perhaps an example

of what could happen to the COVID-19 epidemic going forward Parameterization of each simulation are as follows: a) $\omega_\infty = 2.6 \cdot 10^{-1}$, $k = 4.3$, $\lambda = 137.2$; b) $\omega_\infty = 3.5 \cdot 10^{-1}$, $k = 8.3$, $\lambda = 91.4$; c) $\omega_\infty = 1.3 \cdot 10^{-1}$, $k = 1.8$, $\lambda = 107.2$; and d) $\omega_\infty = 2.2 \cdot 10^{-1}$, $k = 1.1$, $\lambda = 233.2$.
(EPS)

**S11 Fig. Categorical periodic results of simulations.** The same 125,000 as in Fig 4 (10 simulation years) plotted to show other possible results than just annual and sub-annual periodicity. Annual periodicity is defined as any simulation with an inter-epidemic period of between 9 and 18 months and sub-annual of less than 9 months, both requiring more than 5 total waves during the 10 simulation years to filter out strongly damped oscillations. Those with 5 or fewer waves were considered to be damped. Super-annual periodicity is defined as simulations with an inter-epidemic period of greater than 18 months. Because of the possible long-term dynamics of these simulations, damped versus stable periodicity was not determined. Simulations that resulted in a single peak of infections are classified as stable equilibria. The dashed lines represent exponentially waning immunity, a good proxy for the classic SIRS model.
(EPS)

**S12 Fig. COVID-19 annual component and minimum temperature with spline fit.** The same figure as 2B but with spline fits added as well to show the possible non-linear relationship between minimum temperature and epidemic periodicity. Spline fit indicates a possible "threshold" effect, in which regions with very warm winters have very little, if any, annual periodicity. However the sample size with these extreme winter temperatures is small enough that we draw no definitive conclusions besides the more general negative correlative relationship. The lighter blue regression is done with a loess span = 0.5 and the darker blue with the span = 0.8. The linear regression and it's 95% confidence intervals is shown in black and grey.
(EPS)

**S1 Table. Multiple regression of county-level COVID-19 cases to all climatic and demographic variables.** The results of a multivariable linear regression between COVID-19 cases and all numerical variables considered (except for temperature variability and population density as they are combinations of the minimum and maximum temperatures and population size and geographic area respectively). The table includes the coefficients for each variable in the model and their 95% confidence intervals. Variables with significant coefficients are bolded. The entire model is significant with the adjusted $r^2 = 0.541$.
(PDF)

**S2 Table. Parameterization for simulations of SIR model with waning immunity.** The three parameters that govern the waning immunity function were varied based on Latin hypercube sampling. The ranges they were chosen from are given. The other parameters were fixed across all 125,000 simulations.
(PDF)

## Author contributions

**Conceptualization:** Ilan N. Rubin.

**Formal analysis:** Ilan N. Rubin.

**Funding acquisition:** Marc Lipsitch, William P. Hanage.

**Investigation:** Ilan N. Rubin, Marc Lipsitch, William P. Hanage.

**Methodology:** Ilan N. Rubin, Mary Bushman.

**Software:** Ilan N. Rubin, Mary Bushman.

**Supervision:** William P. Hanage.

**Validation:** Ilan N. Rubin.

**Visualization:** Ilan N. Rubin.

**Writing – original draft:** Ilan N. Rubin, Marc Lipsitch, William P. Hanage.

**Writing – review & editing:** Ilan N. Rubin, Marc Lipsitch, William P. Hanage.

## Acknowledgments

We would like to thank colleagues from the Harvard T.H. Chan School of Public Health, Center for Communicable Disease Dynamics for their helpful and thoughtful discussion and feedback throughout the writing of this paper.

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
