## [Editor Report · Decision Letter 0]

21 Oct 2025

PPATHOGENS-D-25-02442

Seasonal forcing and waning immunity drive the sub-annual periodicity of the COVID-19 epidemic

PLOS Pathogens

Dear Dr. Rubin,

Thank you for submitting your manuscript to PLOS Pathogens. After careful consideration, we feel that it has merit but does not fully meet PLOS Pathogens's publication criteria as it currently stands. Therefore, we invite you to submit a revised version of the manuscript that addresses the points raised during the review process.

Please submit your revised manuscript within 60 days Dec 20 2025 11:59PM. If you will need more time than this to complete your revisions, please reply to this message or contact the journal office at plospathogens@plos.org. Please include the following items when submitting your revised manuscript:

We look forward to receiving your revised manuscript.

Kind regards,

Adi Stern

Academic Editor

PLOS Pathogens

Alexander Gorbalenya

Section Editor

PLOS Pathogens Sumita Bhaduri-McIntosh

Editor-in-Chief

PLOS Pathogens

orcid.org/0000-0003-2946-9497 Michael Malim

Editor-in-Chief

PLOS Pathogens

orcid.org/0000-0002-7699-2064

**Additional Editor Comments:**

Dear authors,

I have seen and read your interesting paper together with reviews that were transferred from the journal Science.

I tend to agree very much with the reviewers, that the paper addresses an important and interesting question. However, there are problems both with the modeling and with the data at hand and the ability to answer the opening questions. Could you please revise the paper and add a point-by-point response to the reviewer's comments (some of which seem to have already been addressed).

Additionally, please revisit the text and distinguish the disease (COVID-19) and the virus (SARS-CoV-2). In this regard, statistic inflation of COVID-19 cases by asymptomatic cases of SARS-CoV-2 infection may warrant consideration. You may want to consult https://doi.org/10.1371/journal.pbio.3002130 for the matter. for the matter.

**Journal Requirements:**

https://journals.plos.org/plospathogens/s/submission-guidelines#loc-parts-of-a-submission

3) Your manuscript is missing the following sections: Results, and Discussion.  Please ensure all required sections are present and in the correct order. Make sure section heading levels are clearly indicated in the manuscript text, and limit sub-sections to 3 heading levels. An outline of the required sections can be consulted in our submission guidelines here:

https://journals.plos.org/plospathogens/s/submission-guidelines#loc-parts-of-a-submission

5) We notice that your supplementary Figures, and Tables are included in the manuscript file. Please remove them and upload them with the file type 'Supporting Information'. Please ensure that each Supporting Information file has a legend listed in the manuscript after the references list.

Potential Copyright Issues:

i) Figures 2, and S5. Please (a) provide a direct link to the base layer of the map (i.e., the country or region border shape) and ensure this is also included in the figure legend; and (b) provide a link to the terms of use / license information for the base layer image or shapefile. We cannot publish proprietary or copyrighted maps (e.g. Google Maps, Mapquest) and the terms of use for your map base layer must be compatible with our CC BY 4.0 license.

7) Please amend your detailed Financial Disclosure statement. This is published with the article. It must therefore be completed in full sentences and contain the exact wording you wish to be published.

8) Please send a completed 'Competing Interests' statement, including any COIs declared by your co-authors. If you have no competing interests to declare, please state "The authors have declared that no competing interests exist". Otherwise please declare all competing interests beginning with the statement "I have read the journal's policy and the authors of this manuscript have the following competing interests"

**Reviewers' Comments:**

**Figure resubmission:**

After uploading your figures to PLOS’s NAAS tool - https://ngplosjournals.pagemajik.ai/artanalysis, NAAS will process the files provided and display the results in the "Uploaded Files" section of the page as the processing is complete. If the uploaded figures meet our requirements (or NAAS is able to fix the files to meet our requirements), the figure will be marked as "fixed" above. If NAAS is unable to fix the files, a red "failed" label will appear above. When NAAS has confirmed that the figure files meet our requirements, please download the file via the download option, and include these NAAS processed figure files when submitting your revised manuscript. **Reproducibility:**

---

## [Decision Letter · Decision Letter 1]

3 Feb 2026

PPATHOGENS-D-25-02442R1

Seasonal forcing and waning immunity drive the sub-annual periodicity of the COVID-19 epidemic

PLOS Pathogens

Dear Dr. Rubin,

Thank you for submitting your manuscript to PLOS Pathogens. After careful consideration, we feel that it has merit but does not fully meet PLOS Pathogens's publication criteria as it currently stands. Therefore, we invite you to submit a revised version of the manuscript that addresses the points raised during the review process.

We look forward to receiving your revised manuscript.

Kind regards,

Adi Stern

Academic Editor

PLOS Pathogens

Alexander Gorbalenya

Section Editor

PLOS Pathogens

Sumita Bhaduri-McIntosh

Editor-in-Chief

PLOS Pathogens

orcid.org/0000-0003-2946-9497

Michael Malim

Editor-in-Chief

PLOS Pathogens

orcid.org/0000-0002-7699-2064

**Additional Editor Comments:**

This is a much improved manuscript with valuable insights, and the reviewer suggest some very minor edits here.

Consider replacing "SARS-CoV-2 virus" with "SARS-CoV-2" to avoid redundancy, since "V" stands for "virus"

**Reviewers' Comments:**

Reviewer's Responses to Questions

**Part I - Summary**

Reviewer #1: This is a much-improved manuscript that convincingly argues that the observed patterns in COVID cases across the US between 2020 and 2023 stem from a combination of seasonal forcing and rapid waning of immunity. The additional Discussion text on VoCs and how their emergence and circulation patterns interplay or can be interpreted in the context of this model is helpful.

**Part II – Major Issues: Key Experiments Required for Acceptance**

Please use this section to detail the key new experiments or modifications of existing experiments that should be absolutely required to validate study conclusions.required to validate study conclusions.

Reviewer #1: Model equations (1-5) in the Results section duplicate model equations (9-12) in the Methods section. I suggest removing equations (1-5) and keeping equations (9-12) in the Methods section since the model provides a Method for analysis.

It would be helpful to get an idea of what the annual attack rates are of the subset of simulations that reproduce characteristics of the observed seasonal patterns (e.g., attack rates of simulations using immunity waning functions shown in Fig S7). Are these annual attack rates reasonable/supported by empirical data, e.g., serological surveys/longitudinal studies of antibody dynamics? With such rapid rates of waning and the dynamics shown in the first row of Fig S7, my guess would be that annual attack rates would be relatively high? (Greater than the 10-20% estimated for flu.)

At the state level, how were the min and max temperatures calculated? Were measurements from the most highly populated county used or data somehow averaged across counties (weighted by population size)? Lines 450-452 do not provide sufficient information on this.

**Part III – Minor Issues: Editorial and Data Presentation Modifications**

Reviewer #1: Abstract: please mention time period of COVID studied (the author summary indicates ‘during the pandemic period’ but this is not mentioned in the abstract)

Line 14 (“In the case of the COVID-19 epidemic…”): please include some citations here. If a reader 10 years down the line is reading this paper, these waves might not be in their working memory

Line 47: 7-day rolling average: this is sensible but perhaps include an explanation of why you chose 7-days for the rolling average. (Are there day-of-the-week reporting trends?)

End of Intro: county-level analysis is indicated but large portion of the Results are at the state level. This seems inconsistent. I suggest adding text to the end of the intro section indicating that both state and county level analyses will be performed.

Fig 2B: The linear regression does not seem to capture the data points very well. Consider (only for this panel) to also show a spline fit or logistic-like curve to highlight that there might be a threshold effect?

PLOS authors have the option to publish the peer review history of their article (what does this mean?). If published, this will include your full peer review and any attached files.). If published, this will include your full peer review and any attached files.

.

Reviewer #1: No

**Figure resubmission:**
---

## [Editor Report · Decision Letter 2]

13 Apr 2026

Dear Mr. Rubin,

We are pleased to inform you that your manuscript 'Seasonal forcing and waning immunity drive the sub-annual periodicity of the COVID-19 epidemic' has been provisionally accepted for publication in PLOS Pathogens.

Best regards,

Adi Stern

Academic Editor

PLOS Pathogens

Alexander Gorbalenya

Section Editor

PLOS Pathogens

Sumita Bhaduri-McIntosh

Editor-in-Chief

PLOS Pathogens

orcid.org/0000-0003-2946-9497

Michael Malim

Editor-in-Chief

PLOS Pathogens

orcid.org/0000-0002-7699-2064

All comments have been addressed, looking forward to seeing this out!
---

## [Editor Report · Acceptance letter]

Dear Mr. Rubin,

We are delighted to inform you that your manuscript, "Seasonal forcing and waning immunity drive the sub-annual periodicity of the COVID-19 epidemic," has been formally accepted for publication in PLOS Pathogens.

Best regards,

Sumita Bhaduri-McIntosh

Editor-in-Chief

PLOS Pathogens

orcid.org/0000-0003-2946-9497

Michael Malim

Editor-in-Chief

PLOS Pathogens

orcid.org/0000-0002-7699-2064